# Nanocrystalline Principal Slip Zones and Their Role in Controlling Crustal Fault Rheology

**Berend A. Verberne** [1],*, **Oliver Plümper** [2] and **Christopher J. Spiers** [2]

1 Geological Survey of Japan, National Institute of Advanced Industrial Science and Technology, 1-1-1 Higashi, Tsukuba, Ibaraki 305-8567, Japan
2 Department of Earth Sciences, Utrecht University, P.O. Box 80.021, 3508 TA Utrecht, The Netherlands; o.plumper@uu.nl (O.P.); c.j.spiers@uu.nl (C.J.S.)
* Correspondence: ba.verberne@aist.go.jp; Tel.: +81-29-861-5211

**Abstract:** Principal slip zones (PSZs) are narrow (<10 cm) bands of localized shear deformation that occur in the cores of upper-crustal fault zones where they accommodate the bulk of fault displacement. Natural and experimentally-formed PSZs consistently show the presence of nanocrystallites in the <100 nm size range. Despite the presumed importance of such nanocrystalline (NC) fault rock in controlling fault mechanical behavior, their prevalence and potential role in controlling natural earthquake cycles remains insufficiently investigated. In this contribution, we summarize the physical properties of NC materials that may have a profound effect on fault rheology, and we review the structural characteristics of NC PSZs observed in natural faults and in experiments. Numerous literature reports show that such zones form in a wide range of faulted rock types, under a wide range of conditions pertaining to seismic and a-seismic upper-crustal fault slip, and frequently show an internal crystallographic preferred orientation (CPO) and partial amorphization, as well as forming glossy or "mirror-like" slip surfaces. Given the widespread occurrence of NC PSZs in upper-crustal faults, we suggest that they are of general significance. Specifically, the generally high rates of (diffusion) creep in NC fault rock may play a key role in controlling the depth limits to the seismogenic zone.

**Keywords:** nanograins; principal slip zone; crystallographic preferred orientation; amorphization; mirror-slip surface; faults; earthquakes; localization

## 1. Introduction

Nanocrystalline materials are widespread in the Earth's atmosphere, biosphere, and in the subsurface [1–5], including in principal slip zones (PSZs) within natural faults [6–8]. PSZs are zones of localized shear deformation that (have) accommodate(d) the bulk of displacement in the cores of upper-crustal faults [9,10], which suggests that the physical properties of the ultrafine(nano)-grained fault rock within PSZs plays an important role in controlling fault mechanical behavior or fault rheology. From observations on metals and ceramics it is well known that nanophase materials, characterized by grain sizes < 100 nm, frequently exhibit unusual deformation properties compared with coarser-grained counterparts [11–13]. The reason for this is the loss of cohesive energy between atoms comprising the grain as its size continues to decrease. In view of the generality of this nanograin size effect, it is important to consider the potential physical implications of nanogranular fault rock. Despite the emerging awareness on the importance of nanophase geomaterials in Earth sciences [1–8], their prevalence in upper-crustal faults and potential role in natural earthquake cycles remains insufficiently investigated.

In this paper, we aim to elucidate the significance of nanocrystalline PSZs in Earth's upper crust. We start with background on fault mechanics and upper-crustal seismogenesis, and summarize some key physical properties of nanophase materials which, when applied to fault rock, are expected to be of major importance in controlling fault strength and stability. We go on to review the micro- and nanostructural characteristics of natural and experimentally-formed nanocrystalline PSZs, and list reports from the literature of PSZs characterized by grains <100 nm in size. Our work demonstrates that nanocrystalline PSZs form under a wide range of conditions pertaining slow (a-seismic) and fast (co-seismic) upper-crustal fault slip. Also, we observe that they are frequently characterized by an internal crystallographic preferred orientation, and by the presence of amorphous materials and/or glossy fault plane interfaces known as "mirror-slip" surfaces. Given the abundant observations of nanocrystalline PSZs in field exposures of faults, as well as in experiments, we suggest that they are of general importance to upper-crustal fault deformation. The physical properties of nanocrystalline fault rock may play a key role in natural earthquake cycles, especially in controlling the depth distribution of upper-crustal seismicity.

## 2. Fault Zones, Earthquakes, and the Seismogenic Zone

The presence of long-lived, localized zones of shear deformation in the crust, or fault zones, implies that the fault rocks within are weaker than the surrounding country rocks and that their weakness is persistent [14,15]. The strength of the upper-crust is classically approximated using a Coulomb-type, brittle failure law, abruptly giving way to ductile deformation below ~15 to 20 km depth (Figure 1a) [16,17]. A brittle-to-ductile transition at ~15 to 20 km depth is consistent with geological and seismological observations of the base of the so-called "seismogenic zone", i.e., the depth interval in the upper-crust in which the bulk of upper-crustal earthquakes nucleate [18–25], suggesting that at greater depths earthquake rupture nucleation is inhibited by intrinsically stable, ductile or viscous flow in shear zones. A seismicity cut-off at shallower depths, typically observed at ~2–4 km, demarcates the upper limit of the seismogenic zone [24,26]. Field and laboratory studies of fault deformation suggest that within the seismogenic zone, "multi-mechanism" or "frictional-viscous" fault slip-involving coincident rate-sensitive (creep) and rate-insensitive (e.g., cataclasis) deformation mechanisms-plays an important role (Figure 1b) [27–37]. However, in general, the microphysical processes responsible for aseismic fault sliding above the seismogenic zone, and for seismogenic slip within, remain poorly understood for most fault rock types.

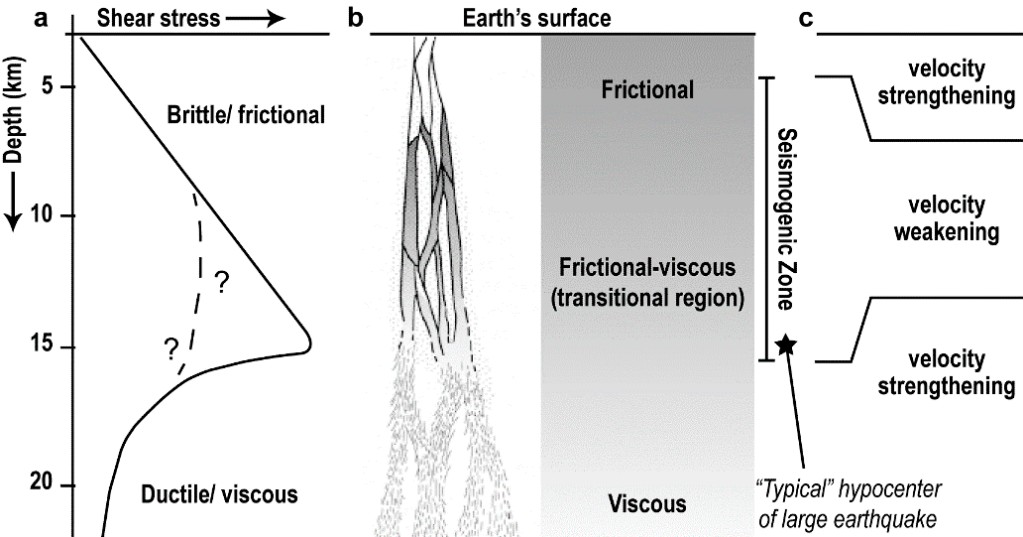

**Figure 1.** Schematic profile of a fault zone in Earth's upper-crust. (**a**) Fault strength vs. depth. (**b**) Fault zone sketch (from [15]), with a rough depth range indicating fault deformation regimes, the seismogenic zone, and (**c**) velocity dependence or intrinsic fault stability regimes.

In the case of earthquakes, sliding along faults is achieved by unstable, periodic slip events instead of by stable, continuous motion. This is similar to the jerky sliding motion that is frequently observed in laboratory rock friction experiments, known as "stick-slip" [38]. Regular stick-slip behavior can be easily envisioned using a spring-block model system, consisting of a rigid block or slider on a nominally flat surface, driven via a spring of a certain stiffness. When the spring is pulled at constant speed, an instability may develop depending on the frictional properties of the slider-surface contact, the mass of the block, the spring stiffness, and the loading rate, resulting in intermittent slider acceleration and stationary contact [39–42]. Ruina [41] showed that for regular stick-slip to occur the slider-surface contact must decrease in strength with increasing displacement rate, hence be "velocity-weakening". In the opposite case "velocity-strengthening" occurs, which leads to a state of stable sliding [41,43]. Thus, the seismogenic zone is believed to represent a depth interval in the upper-crust where shear deformation of fault zones leads to unstable, velocity-weakening behavior, as opposed to stable velocity-strengthening above and below (Figure 1c) [22,24,25,44].

Importantly, the velocity dependence of frictional strength is a material property of the sliding medium which constitutes the slider-surface contact. Applying this to natural faults, the sliding medium is represented by the granular wear product of cumulative slip along the fault, or "fault gouge", present in the fault core [45]. Field and drilling studies of active and inactive natural fault zones frequently demonstrate the presence of a mm- to cm-wide principal slip zone (PSZ) in the gouge-filled fault core that accommodates, or has accommodated, the bulk of displacement along the fault [9,46–52]. Tectonic loading of a faulted rock volume, as occurs continuously in numerous geologic settings (e.g., at tectonic plate boundaries), causes energy dissipation predominantly along the PSZ [10,53]. In the case of slow (aseismic) fault sliding, quasi-static deformation of fault rock is believed to be key [54–56], whereas at higher (seismic) slip rates frictional heat generated along the PSZ plays an increasingly important role [57], leading to dynamic fault rupture processes such as melting, decarbonation, and/or thermal pressurization [58–63].

## 3. The Physical Properties of Nanophase Materials

Material properties such as melting temperature or yield point frequently show drastic changes when the grain size decreases to the nanometer-realm (<100 nm) [11–13]. The reason for this is fundamental; a decreasing grain size implies a parabolic increase of the fraction of surface atoms (Figure 2a), which have a much lower average binding energy compared with atoms in the bulk phase. This means that when the grain size decreases to that of a few atoms or unit cells, it has major implications for thermodynamic stability and reactivity of the individual particles [64–66]. For example, the melting point of Au particles is observed to decrease from ~1300 K to 700 K as the grain size decreases from 20 nm to 5 nm [67] (Figure 2b). Observations on common rock-forming minerals are scarcer. However, in the case of calcite, which is the dominant constituent of limestone, the decomposition temperature decreases from ~1075 K to 950 K as the grain size decreases from 40 nm to 20 nm [68] (Figure 2b). Size-dependence of the melting or decomposition temperature of fault rock within a principal slip zone (PSZ) may have major implications for bulk fault rheology, for example at elevated temperatures due to frictional heating.

Another unique aspect relevant to nanostructured polycrystals and fault rock is their huge cumulative grain surface area, which naturally increases exponentially as the grain size continues to decrease (Figure 2c). This has major implications not only for chemical reactivity but also for the rheology of a material. For example, due to the short grain scale transport distances in nanostructured materials [69], grain boundary diffusion driven mechanisms [70–72] are generally fast, enabling superplastic deformation at much lower temperatures/higher strain rates than compared with in coarser-grained materials [73–75]. The high grain boundary density also plays an important role in dislocation-mediated plasticity. As the grain size decreases to the <100 nm size range, dislocations are emitted and adsorbed efficiently at grain boundaries, leading to a decrease of the material yield strength with decreasing grain size hence an "inverse Hall-Petch effect" [76–79] (Figure 2d). In this

mechanism, dislocations traverse the crystallite within very brief time windows, achieving very large strains while leaving a micro-/nanostructure characterized by "strain-free" nanograins [80–82].

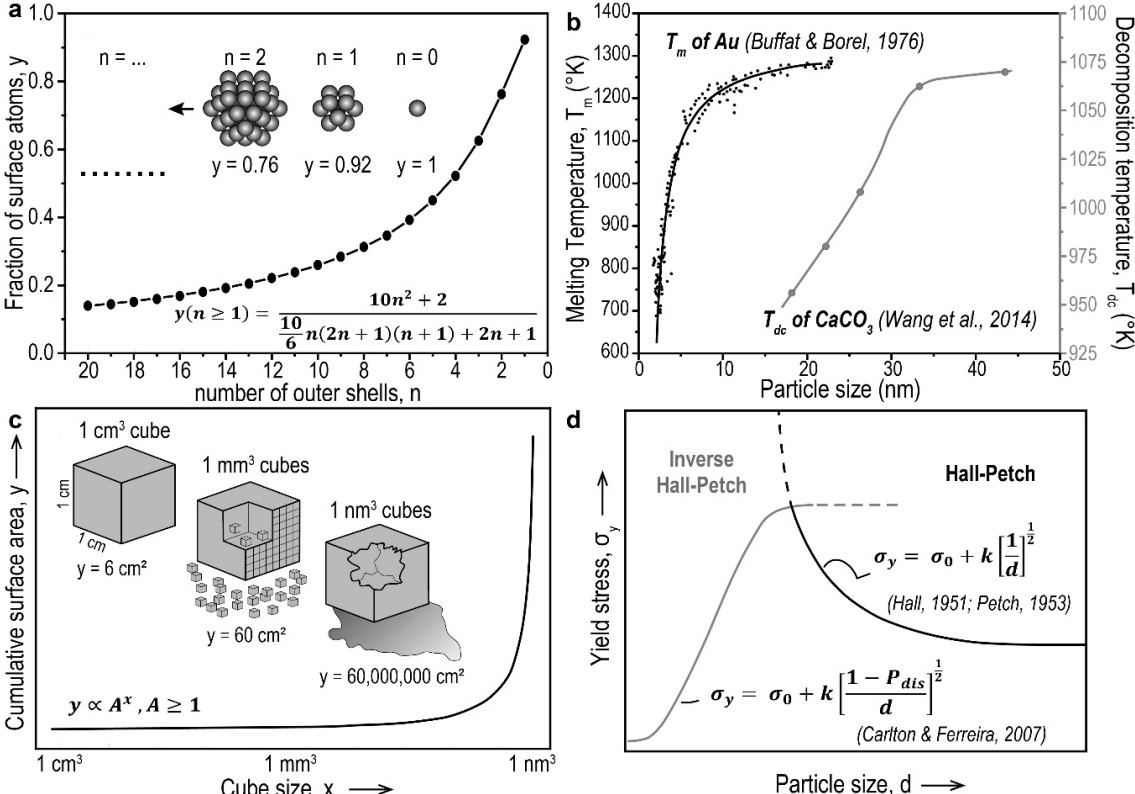

**Figure 2.** Key characteristics of nanoparticles and nanostructured materials. (**a**) A simple model of hexagonal close-packed balls illustrating the effect of particle miniaturization. The fraction of surface particles increases near-exponentially with decreasing number of outer shells, or particle size (after [11,13]). (**b**) The melting temperature of Au [67] and the decomposition temperature of $CaCO_3$ [68] particles decrease sharply as particle size decreases within the nm-realm. (**c**) The cumulative surface area of polycrystals increases exponentially as the grain size continues to decrease. (**d**) The empirical relation between yield strength $\sigma_y$ and grain size $d$, known as the Hall-Petch (HP) effect [76,77], reverses for very small $d$ (after [78,79]). The expression for $\sigma_y$ in the inverse-HP regime is the model by Carlton & Ferreira (2007) [79], where $k$ is a constant and $P_{dis}$ is the probability of a dislocation being absorbed by a grain boundary.

## 4. Nanocrystalline Principal Slip Zones in Natural Faults and in Experiments

Observations of natural and experimentally-formed principal slip zones (PSZs) showing the presence of <100 nm-sized grains are listed in respectively Tables 1 and 2. Below we summarize the micro- and nanostructural characteristics of nanocrystalline PSZs, highlighting seminal reports of field/drilling studies of natural faults and of laboratory studies. Much insight was obtained recently from studies of glossy or "mirror-like" fault slip surfaces (MSSs) formed in experimentally simulated faults composed of calcite fault gouge. These are described using a separate section. While we aspire to provide as complete an overview as possible, we may have overlooked some of the studies performed to date.

### 4.1. Nanocrystalline Principal Slip Zones in Exposures of Natural Faults

Faults that are exposed in an orientation normal to the fault plane display a cross-section through the damage zone that has developed upon repeated fault displacement, including the principal slip zone(s) (Figure 3a). Power and Tullis [83] used optical and transmission electron microscopy (TEM)

to investigate sections prepared normal to the fault plane of rocks collected from the glossy fault trace of the Dixie Valley thrust fault (USA). In a zone just ~0.2 mm wide, the fault trace or PSZ is characterized by ultrafine grains down to 10 nm in size, and a uniform optical birefringence and extinction. This optical effect may be observed in (ultra)thin sections using crossed nicols in a polarizing light microscope (by rotating the microscope stage as shown in Supplementary Video S1) and is widely used as indicative of a crystallographic preferred orientation (CPO). Chester & Goldsby [84] also reported a nanocrystalline PSZ with a CPO, in fault core samples from the Punchbowl Fault (USA). Field investigation revealed visually distinct, 0.15 to 0.55 m thick layers of fine-grained fault gouge known as ultracataclasite, separated by what was identified as a "principal fracture surface" [46]. However, thin section analyses revealed that the ultracataclasite layers were separated by a zone of finite width (constituting a principal slip *zone*), characterized by a strong uniform birefringence, and the presence of grains down to 4 nm in size [7,84]. Other notable observations of naturally-formed nanocrystalline PSZs have been made from drilling of seismically-active fault zones, such as the Chelungpu Fault (Taiwan) [8,85] and the San Andreas Fault (USA) [86].

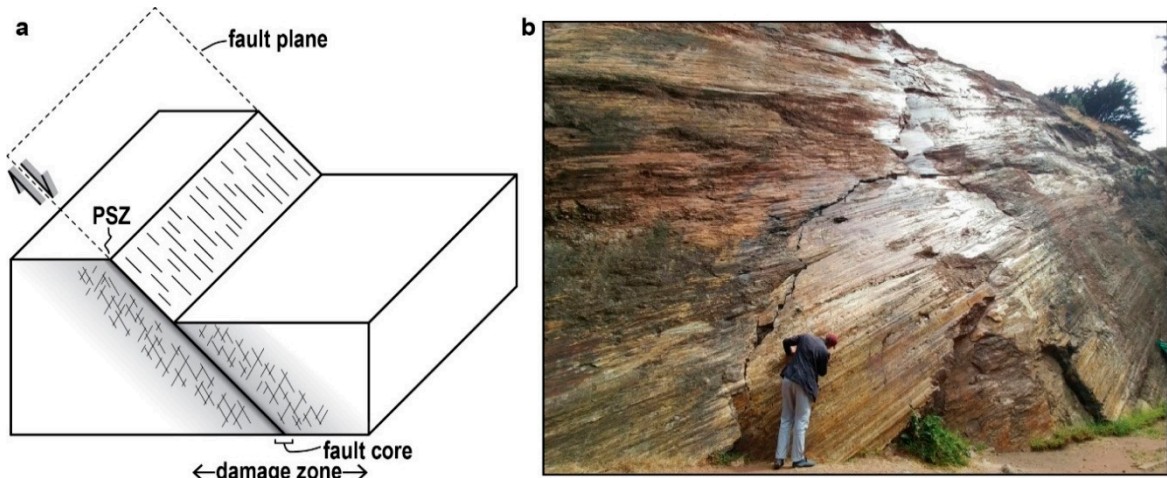

**Figure 3.** Principal slip zones (PSZs) in natural faults. (**a**) Sketch of a normal fault, highlighting the fault plane, damage zone, fault core, and PSZ (after [10,47]). (**b**). Striated, glossy surface of the Corona Heights Fault (USA) (courtesy of J. E. Samuelson).

Faults that are exposed parallel to the fault plane display the fault core, which is frequently characterized by slip-parallel striations and a relatively erosion-resistant, "glossy" or "well-polished" surface (Figure 3b) (Table 1). Such exposures have been reported as meter-scale outcrops in the field [87, 88], but also as cm-scale patches in drill core samples of active faults [85,86]. Siman-Tov et al. [87] coined the term "fault mirrors" for highly light-reflective fault surfaces cutting carbonate rocks in the Dead Sea transform region (Israel). They showed that the glossy fault plane is internally composed of a thin (<1 μm) veneer of calcite grains of a size down to ~50 nm. However, a glossy or mirror-like appearance has been described for fault surfaces composed of 0.1–1 μm-sized grains [85,89,90], and do not reveal much about the grain size within. As pointed out by Siman-Tov et al., the specular reflectivity occurs because the fault surface roughness has a wavelength shorter than that of visible light (400 nm) [91].

Another notable observation that has been frequently reported on using samples from naturally occurring nanocrystalline PSZs is the presence of (partly) amorphized material (Table 1). It may be observed as cm-thick veins in the field [92], or as thin coatings surrounding mineral clasts when observed using TEM [86,93,94]. Veins of glassy, amorphized rock known as pseudotachylytes, may form as a result of frictional melting along faults, pointing to high (co-seismic) slip rates [45,95,96]. For this reason, melt-origin pseudotachylytes are frequently used as field indicators of paleo-earthquake rupture [97,98]. However, the formation process is not implicit to the definition of pseudotachylytes,

and they may form by other mechanisms than seismically-induced frictional melting [99]. Caution is necessary on the interpretation of field exposures of faults showing the presence of amorphous veins.

**Table 1.** List of reports, in chronological order, of ≤100 nm-sized grains in the cores (principal slip zones) of natural faults.

| Location | Dominant Host Rock Mineralogy | $d$ (nm) | CPO? | Glossy Surface? | Amorphous Material? | Source |
|---|---|---|---|---|---|---|
| Dixie Valley Thrust, USA | quartz | 10 | √ | √ | | Power & Tullis [83] |
| Punchbowl Fault, USA | quartz, feldspar, clays | 4 | √ | | | Chester & Goldsby [84] |
| Chelungpu Fault (TCDP borehole C) | quartz, clays | 50 | | | | Ma et al. [8] |
| Nojima Fault Zone, Japan | quartz, feldspar | 30 | | | | Keulen et al. [100] |
| Iida-Matsukawa Fault, Japan | quartz, feldspar | 20 | | | √ | Ozawa & Takizawa [92] |
| San Andreas Fault (SAFOD main hole) | clays, quartz, feldspar | 50 | | √ | √ | Janssen et al. [86] |
| Kfar Gladi Fault, Israel | calcite | 50 | | √ | | Siman-Tov et al. [87] |
| Corona Heights Fault, USA | silica, quartz | 10 | √ | √ | √ | Kirkpatrick et al. [88] |
| Gubbio Fault, Italy | calcite, clays | 50 | | √ | | Bullock et al. [101] |
| Mt. Maggio fault, Italy | calcite | 100 | | | | Collettini et al. [102] |
| Vado di Corno fault, Italy | calcite, dolomite | 50 | | √ | | Demurtas et al. [103] |
| Capolivieri-Porto Azurro shear zone, Italy | tourmaline | 10–100 [†] | | | √ | Viti et al. [90] |
| Hsiaotungshi fault system (borehole), Taiwan | quartz, clays | 50–100 | | √ | √ | Kuo et al. [85] |
| Maclure Glacier, USA | feldspar, quartz | 10–100 | | √ | √ | Siman-Tov et al. [104] |
| Mt. Vettore Fault, Italy | calcite, clays | 50 | | √ | | Smeraglia et al. [105] |

"*d*" is the minimum grain size observed. [†] Fault mirrors (glossy surfaces) are also reported, however here they are composed mainly of >200 nm sized crystals. See the respective papers for details on localities, outcrops, and PSZ formation conditions.

### 4.2. Nanocrystalline Principal Slip Zones Formed in Fault-Slip Experiments

Laboratory experiments aiming to investigate upper-crustal fault deformation are carried out by imposing displacement along (initially) bare rock surfaces, or on a powdered sample layer representing a simulated fault gouge. The technology used to conduct fault-slip experiments varies greatly [106–108]. However, for simplicity, here we distinguish between two types of fault-slip tests; (i) low-velocity friction (LVF) tests, used to study slow fault-slip including the early (nucleation) stages of earthquake rupture, and (ii) high-velocity friction (HVF) tests, used to study dynamic earthquake rupture processes. Following Rowe & Griffith [98], slip rates ($v$) beyond ~$10^{-4}$ m/s are "almost certainly dynamic", so we define HVF tests as using $v \geq 10^{-4}$ m/s, and LVF tests as using $v < 10^{-4}$ m/s. There are numerous other differences between LVF and HVF tests, in addition to the displacement rate used, that may affect micro- and nanostructural development along the simulated fault, or its recovery after an experiment. LVF tests typically achieve steady-state conditions of normal stress ($\sigma_n$) and temperature ($T$), but reaching cumulative displacements of maximally a few centimeters ($\sum x = 10^{-3}$–$10^{-2}$ m). HVF tests may run for meters ($\sum x = 10^{1}$–$10^{2}$ m), with frictional heat generated at the fault-slip interface playing an important role.

Despite the major differences between LVF and HVF tests, simulated fault samples recovered after an experiment typically show one or more, ultrafine-grained, shear plane-parallel bands of finite width, located in the sample interior ("Y-shears") or close to the loading piston interface ("boundary shears") [109,110] (Figure 4a–d). These shear bands mark a zone of abrupt grain size reduction with respect to the host rock (Figure 4b,d), and accommodated the bulk of the imposed shear displacement, i.e., representing experimentally-formed principal slip zones (PSZs). PSZ thicknesses may range from ~50–100 μm in samples recovered from LVF tests, to a few (tens of) microns in HVF deformed samples. Yund et al. [111] used TEM to investigate PSZs developed in simulated fault gouges of siliceous and carbonate compositions deformed in LVF and HVF rotary shear tests (Table 2). They reported grain sizes down to ~10–50 nm in all samples investigated, and the presence of amorphized materials, except in the carbonates. However, there are numerous (recent) reports of nanocrystalline PSZs formed in

LVF and HVF experiments using simulated fault samples composed of carbonates [58,93,112–119], many of which also showed the presence of amorphous materials (Table 2).

A crystallographic preferred orientation (CPO) was reported for nanocrystalline PSZs developed in LVF tests using simulated gouges composed of calcite [113,114,118] and quartz [120] (Table 2). The presence of a CPO may be inferred from uniform birefringence and extinction observed in thin sections (Supplementary Video S1), or else demonstrated using selected area diffraction in TEM. Electron backscatter diffraction is a powerful tool frequently used to quantify a CPO [121], however, to our knowledge this remains difficult to apply to extremely fine-grained aggregates such as those characterizing nanocrystalline PSZs (grain size << 100 nm). Recently, EBSD measurements were used to quantify a CPO characterizing a PSZ composed of 200–300 nm-sized, polygonal grains, formed in simulated calcite gouge sheared in HVF tests [122,123].

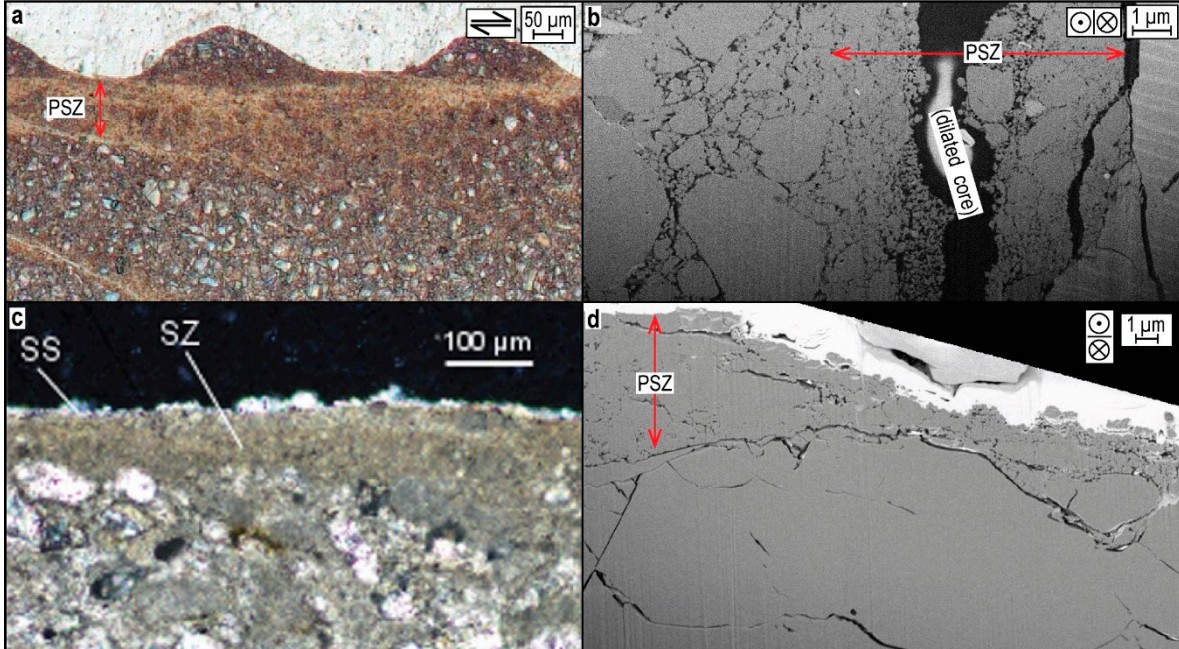

**Figure 4.** Principal slip zones (PSZs) in simulated calcite(-rich) fault gouge formed in LVF (**a**,**b**) and HVF experiments (**c**,**d**). (**a**) Plane polarized light micrograph of an ultra-thin section (parallel to the slip vector, sample CaCO$_3$-RT-dry of [113]). (**b**) Backscatter electron (BSE) micrograph prepared using a focused ion beam scanning electron microscope (FIB-SEM) (normal to the slip vector, sample lmst@150 °C of [124]). The central void is due to post-test dilation. (**c**) Cross-polarized light micrograph showing a narrow, fine-grained slip zone (SZ) bound by a slip surface (SS). Taken with publisher's permission from [93]. (**d**) BSE micrograph. Taken from [115].

**Table 2.** List of reports, in chronological order, of <100 nm-sized grains in experimentally-formed principal slip zones.

| Dominant Sample Mineralogy | $v_{max}$ (m/s) | $\sigma_n$ (MPa) | $\sum x$ (m) | $d$ (nm) | CPO? | Glossy Surface(s)? | Amorphous Material? | Source |
|---|---|---|---|---|---|---|---|---|
| quartz | $10^{-5}$ | ~135 | $10^{-5}$ | 90 | | | √[†] | Engelder [125] |
| quartz | $10^{-2.5}$ | 7–50 | $10^{-1}$ | 10–15 | | | √ | Yund et al. [111] |
| quartz, feldspar | $10^{-2.5}$–$10^{-6}$ | 50–75 | $10^{-2}$–$10^{-1}$ | 10–15 | | | √ | Yund et al. [111] |
| calcite | $10^{-2.5}$ | 15 | $10^{-2}$ | ~50 | | | | Yund et al. [111] |
| dolomite | $10^{-2.5}$ | 75 | $10^{-1}$ | ~50 | | | | Yund et al. [111] |
| calcite | $10^{-2}$–$10^{0}$ | 1.1–13.4 | $10^{0}$–$10^{1}$ | 10 | | √ | | Han et al. [58] |
| siderite, magnetite | $10^{0}$ | 0.6–1.3 | $10^{1}$ | 20–30 | | √ | | Han et al. [112] |
| antigorite | $10^{0}$ | 24.5 | $10^{0}$ | ~50 | | | √ | Viti & Hirose [126] |
| quartz, feldspar | $10^{-7}$–$10^{-8}$ | >$10^{3}$ | $10^{-3}$ | 8 | | √ | √ | Pec et al. [127,128] |
| calcite | $10^{-7}$–$10^{-5}$ | 50 | $10^{-3}$ | 5 | √ | √ | √[††] | Verberne et al. [113,114] |
| dolomite | $10^{0}$ | 28.4 | $10^{-1}$ | 10 | | √ | | Green II et al. [116] |
| quartz, feldspar | $10^{-6}$–$10^{-5}$ | 25 | $10^{-2}$–$10^{-1}$ | 15–50 | | | √ | Hadizadeh et al. [129] |
| quartz, clays | $10^{0}$ | 1 | $10^{1}$ | 10–50 | | | √ | Kuo et al. [130] |
| quartz, silica | $10^{-6}$ | >$10^{3}$[†] | $10^{-3}$ | 5 | √ | | √ | Toy et al. [120] |
| calcite | $10^{-1}$–$10^{0}$ | 10 | $10^{-3}$–$10^{1}$ | 5–10 | | √ | √ | Spagnuolo et al. [115] |
| calcite | $10^{-1}$ | 0.47–1.57 | $10^{1}$ | 45 | | √ | | Siman−Tov et al. [117] |
| quartz | $10^{-7}$–$10^{-3}$[‡] | 92–287 | $10^{-3}$ | 10 | | √ | √ | Hayward et al. [131] |
| quartz, smectite | $10^{-4}$–$10^{0}$ | 5 | $10^{0}$ | 10–50 | | | √ | Aretusini et al. [132] |
| quartz, muscovite | $10^{-8}$–$10^{-5}$ | 120 | $10^{-4}$ | 10 | √ | | | Niemeijer [133] |
| quartz, muscovite | $10^{-4}$ | 120 | $10^{-4}$ | 10 | √ | | | Niemeijer [133] |
| calcite | $10^{-7}$–$10^{-5}$ | 20–100 | $10^{-3}$–$10^{-4}$ | 10 | | | | Mercuri et al. [119] |
| calcite | $10^{-6}$–$10^{-5}$ | 45 | $10^{-1}$–$10^{-2}$ | 50 | √ | | √ | Delle Piane et al. [118] |
| quartz | $10^{-4}$–$10^{-1}$ | 2.5–5 | $10^{0}$–$10^{1}$ | 10 | | √ | √ | Rowe et al. [134] |

Grey shaded rows include data from LVF tests (here defined as tests employing $v < 10^{-4}$ m/s). $v_{max}$ = max. displacement rate; $\sigma_n$ = (effective) normal stress; $\sum x$ = accumulated displacement; $d$ = min. grain size. Only the orders of magnitude of $v_{max}$ and $\sum x$ are given. Glossy surfaces refer to the presence of "shiny" or "mirror-like" surfaces, regardless of continuity. For details see the respective literature. [†] Attributed to beam damage. [††] Attributed to unknown contamination. [‡] $v_{max}$ here was estimated from unstable slip events.

Mirror-Slip Surfaces in Principal Slip Zones Developed in Calcite Gouge

"Glossy", "shiny", or "mirror-like" slip surfaces (MSSs) have been observed in experiments using a wide range of sample materials, characterized by a wide range of normal stresses ($\sigma_n$), displacement rates ($v$), and cumulative displacements ($\sum x$) (Table 2) (Figures 5 and 6). Recently, much attention has been given to MSSs developed in simulated faults composed of calcite, mainly because of their striking similarity with carbonate "fault mirrors" frequently observed in tectonically-active carbonate terrains [87,135], and the question whether they may be indicators of past seismic slip.

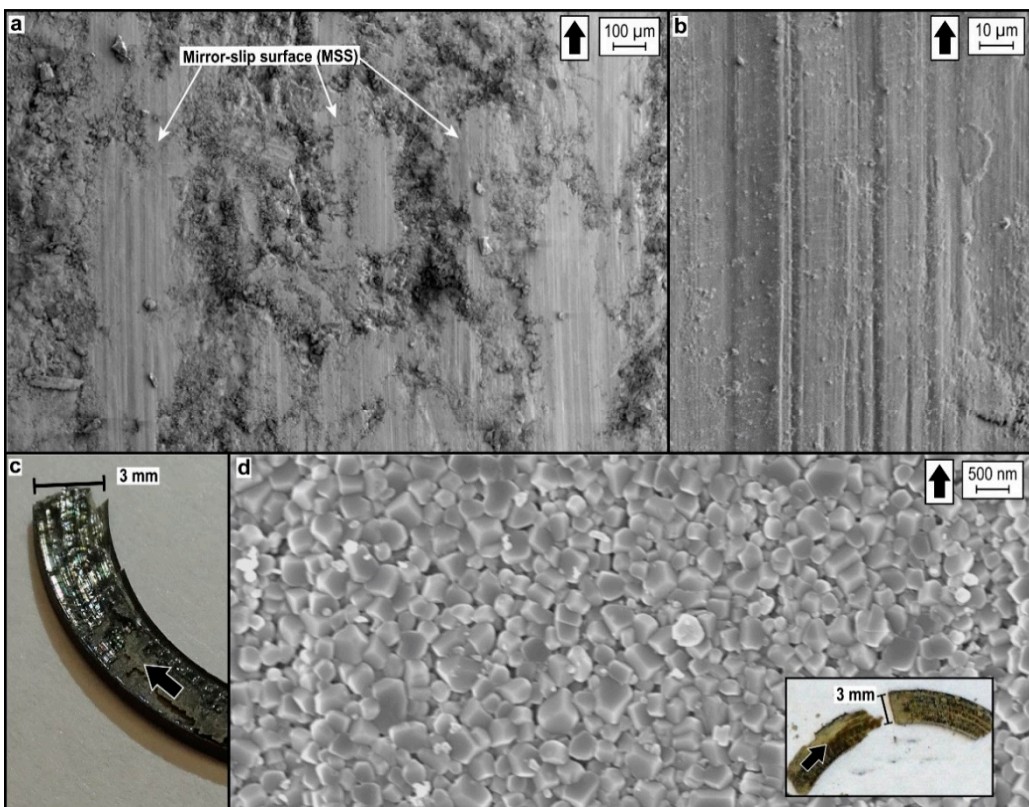

**Figure 5.** Principal slip zones with mirror-slip surfaces formed in simulated calcite gouge sheared in HVF tests ($v \geq 10^{-4}$ ms$^{-1}$). (**a**,**b**) Secondary electron (SE) micrographs. From [115]. (**c**) Sample fragment recovered from an experiment conducted using $0.1 \leq v \leq 100$ μms$^{-1}$, $\sigma_n^{eff} = 50$ MPa, $T = 550$ °C, $\sum x = 24.7$ mm (unpublished data). (**d**) Top view onto the PSZ developed in an experiment conducted at $v = 100$ μms$^{-1}$ (sample CaCO$_3$-550-vhigh of [122]). Inset shows a photo of the (fragmented) sample recovered after the experiment.

In general, MSSs are characterized by extremely low surface roughness, especially in a direction parallel to the shear direction [136,137]. They have been reported as multiple, elongated patches aligned parallel to the shear direction (Figures 5a and 6a), or else as a single, continuous interface marking the PSZ boundary (Figure 5c). The number and extent of MSSs were shown to increase with increasing displacement and/or displacement rates, in HVF tests conducted at normal stresses up to 26 MPa [117,135,138], which led some authors to conclude that continuous MSSs may indeed serve as indicators of past seismic slip in natural faults cutting carbonates. However, a continuous MSS has also been observed in simulated calcite gouge sheared at $v = 10$ μms$^{-1}$ (effective normal stress 20 MPa $\leq \sigma_n^{eff} \leq 100$ MPa, $T \approx 550$ °C, $\sum x = 12.4$ mm; Figure S2D of [122]). The role of effective normal stress, and of cumulative displacement, on the formation of (patchy) MSSs with progressive shear strain in LVF tests remains to be investigated. Pozzi et al. [139] recently reported on the microstructural development of MSS-bearing PSZs with progressive shear strain in HVF experiments on simulated calcite faults ($\sigma_n = 25$ MPa, $v$ up to 1.4 m/s). Using polished sections prepared normal to the slip vector

they observed that, after an initial stage of slip ($\Sigma x \approx 7$ cm), sharp discontinuities develop which are interpreted to represent MSSs. The matured PSZ is observed to consist of 200–300 nm sized, polygonal grains characterized by a crystallographic preferred orientation [123].

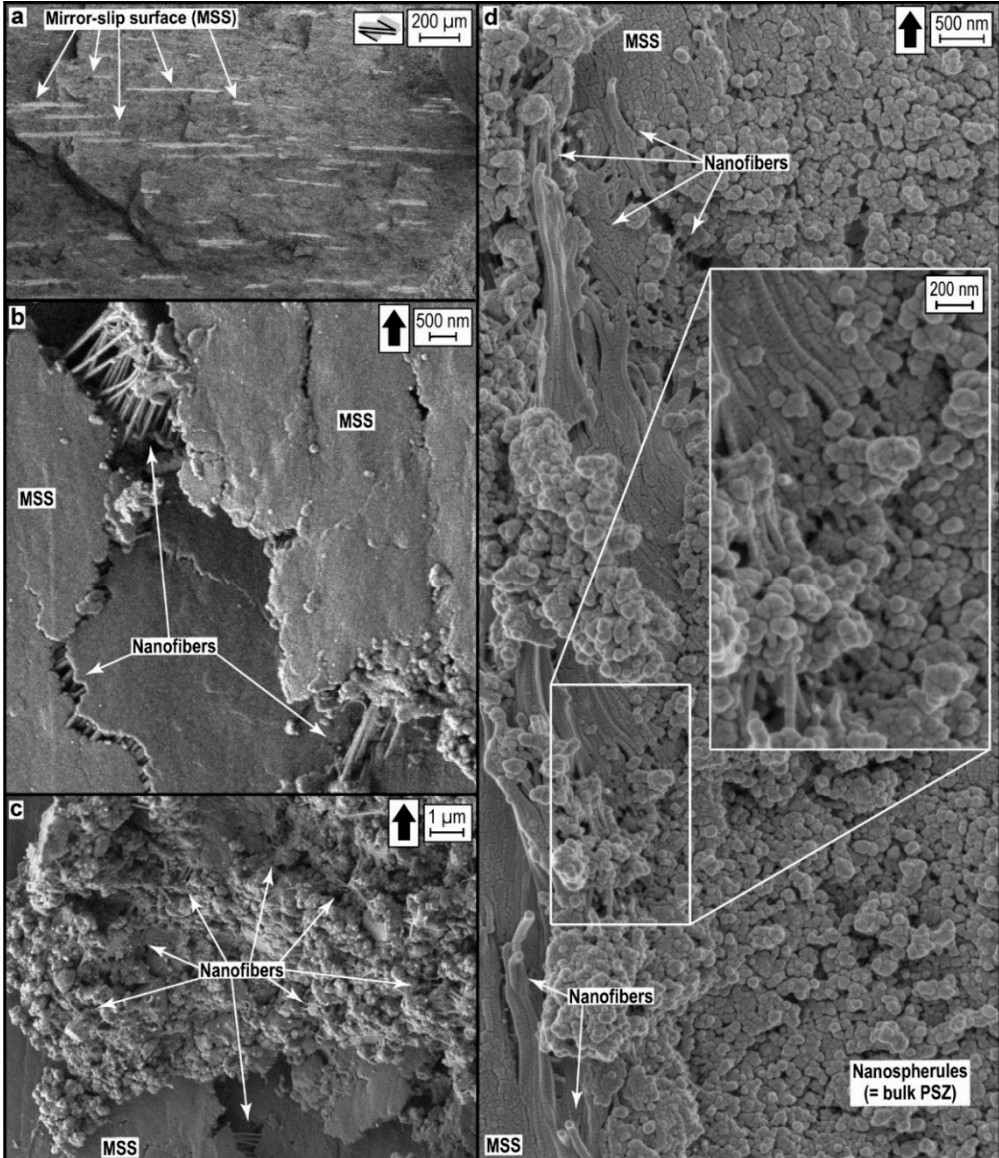

**Figure 6.** Principal slip zone with mirror-slip surfaces formed in simulated calcite gouge sheared in LVF tests ($v = 10^{-6}$ to $10^{-5}$ ms$^{-1}$, see [140]). Secondary electron micrographs. (**a**) Taken at an angle of 52° to the shear plane. The patches that are elongated parallel to the shear direction represent MSSs. (**b**) Stretched nanofibers and (**c**) nanofibers within the bulk PSZ. (**d**) Alignment of nanospherules at the edge of an MSS. (a) to (d) are top views onto the shear plane. The micrographs shown in (**a**,**d**) are from sample SEMB of [140], taken using a FEI Nova Nanolab FIB-SEM. The micrographs in (**b**,**c**) are of samples sheared using a gas-medium deformation apparatus installed at the Geological Survey of Japan (Tsukuba, Japan), and taken using a JEOL-7400F FEG-SEM.

Returning to patchy MSSs formed in LVF experiments using calcite gouge ($v = 10^{-6}$ m/s, $\sigma_n{}^{eff} = 50$ MPa, $\sum x \approx 5$–6 mm), individually these show remarkable micro- and nanostructural characteristics (Figure 6a–d) [140]. The PSZ itself comprises a porous, sheet-like volume of ~100 nm-sized spherical particles, with internal, 0.1 to 1 μm-thick, dense planar coatings comprising the MSSs. The MSS patches are observed at different topographic levels within the PSZ (Figure 6a) and are internally

composed of ~100 nm wide fibers that show marked extension and plastic bending when stretched (Figure 6b–d). At locations where stretching led to nanofiber failure, necking structures are absent, suggestive of a low stress-sensitivity of the ductile strain rate (a low "*n*-value"), or superplastic behavior [140]. Nanofiber stretching in this way could only have occurred upon opening of the microcracks at room conditions after the experiment. The nanofibers are locally observed away from MSSs, i.e., within the porous volume constituting the broader PSZ (Figure 6c). Selected area diffraction patterns, taken using TEM, of a single nanofiber as well as of the spherical particles comprising the bulk PSZ, revealed a polycrystalline substructure composed of crystallites 5 to 20 nm in size, characterized by a CPO [113,114,140]. The uniform width of the nanofibers and the spherical nanograin aggregates or nanospherules, (both with diameter ~100 nm), combined with the alignment observed of nanospherules at the edge of some MSSs (Figure 6d), suggest that the nanofibers represent linear nanospherule chains.

## 5. Discussion

The compilation of literature observations reported above (Tables 1 and 2) demonstrates that nanocrystalline principal slip zones (NC PSZs) form in a wide range of rock types under a wide range of normal stresses and displacement rates pertaining to co-seismic and sub-seismic fault-slip in Earth's upper-crust. This suggests that NC PSZs play an important role in controlling fault sliding behavior, including earthquake rupture nucleation and dynamic propagation. Below we discuss possible formation mechanisms of the PSZ nanostructures observed, as well as a comparison between mirror-slip-surface-bearing PSZs developed in low-velocity friction (LVF; $v < 10^{-4}$ ms$^{-1}$) and in high-velocity friction (HVF; $v \geq 10^{-4}$ ms$^{-1}$) tests. We go on to discuss the role of NC PSZs in controlling upper-crustal fault strength and stability, and we consider their broader significance in the seismogenic zone.

### 5.1. Formation of PSZ Nanostructures, Amorphous Materials, and CPO

Under brittle conditions in a fault zone, grain size reduction occurs by cataclastic deformation involving intragranular fracture, comminution and intergranular friction [100,141–143]. However, below a certain critical grain size $d_{\text{crit}}$ known as the grind limit, the stress required to initiate a fracture in compression becomes too high so that plastic yielding occurs [144]. For quartz, $d_{\text{crit}} \approx 0.9$ µm [100], whereas for calcite $d_{\text{crit}} \approx 0.85$ µm [145]. This means that the <100 nm-sized grains frequently observed in fault-slip experiments must point to a mechanism of grain size reduction involving plastic yielding. Using a model based on mode I Griffith failure [144,146] and low-temperature plasticity, Sammis and Ben-Zion [147] showed that in the case of quartz in a compressive regime, shock loading and subcritical crack growth may produce particles down to 3 nm in size. However, specifically for LVF experiments which employ displacement rates of µm/s and reach just millimeters of cumulative displacement over the timespan of a few hours, the formation of crystallites down to 5 nm in size combined with the presence of amorphous materials and a crystallographic preferred orientation (CPO) (Table 2), remains intriguing.

Focusing on simulated calcite faults (Figures 4–6), the internal polycrystalline substructure observed in PSZs nanograins formed in LVF [113,114,140] as well as HVF experiments [93] bears a striking similarity to microstructures found in shocked ductile metals [148,149]. As in metals, the high ductility of calcite [150,151] may therefore allow the observed ~5–20 nm substructure to form by progressive development of nano-cell walls from dense dislocation networks and tangles generated by low temperature crystal-plasticity (e.g., r(104) slip or e(108) twinning [152]). Following from this, we speculate that plastic deformation and/or fracturing and abrasion occurring at parent grain surfaces led to the detachment of ~100 nm sized nanocrystalline clusters or fragments from these micron-sized parent grains [148,149]. The nanograins produced in turn rounded, to form the rolling, grain-neighbor-swapping nanospherules comprising the porous nanogranular PSZ [140]. To further unravel the formation mechanism of nanocrystallites and nanospherules in calcite gouge is a challenging task which requires more elaborate experiments and micro-/nanostructural analyses.

Amorphous materials may form by melt quenching, mechanical deformation, chemical reactions or a coupling between the latter two. Here, we focus on those materials derived from sub-solidus derived processes. For a thought-provoking investigation into chemo-mechanical- vs. melt-derived amorphous solids the reader is referred to Pec et al. [127]. In general, solid-state amorphization is attributed to arise from (1) the introduction of externally-derived mechanical instabilities (e.g., dislocations), (2) externally-forced volume expansion at constant temperature or (3) thermal expansion during heating at constant pressure [65,153]. By contrast, mechano-chemical interactions may produce amorphous materials as a result of the reduction of chemical species. This is particularly relevant in carbonate fault rocks where decarbonation reactions release $CO_2$ that can subsequently be reduced to (amorphous) carbon phases. Several natural [154–156] and experimental [93,115,118,140] studies of carbonate fault rocks have reported the occurs of amorphous carbon, often intimately associated with nanogranular calcites. The exact chemical pathways for $CO_2$ reduction to amorphous carbon remain, however, debated [115,156,157]. Additionally, in natural systems fluids can facilitate the precipitation of amorphous solids [158], making it difficult to discriminate internally- from externally-controlled formation mechanisms. Sub-solidus-derived amorphous materials are also widely reported in silicate-dominated systems (Table 2). The detailed mechanism(s) of amorphization, and more generally the impact of differences in atomic order on mechanical properties [159], on fault rheology, remains subject of further study.

Aside from the formation of nanocrystallites and amorphization within a PSZ, this does not explain the development of an internal CPO. Pozzi et al. [123] suggested that grain-size insensitive (GSI) creep mechanisms (dislocation creep) may explain CPO formation in a PSZ composed of 200–300 nm-sized grains formed in calcite gouge sheared at $v$ = 1.4 m/s, at $\sigma_n$ = 25 MPa. However, in the case of CPO-bearing PSZs formed in LVF tests, the porous structure observed in the bulk PSZ (Figure 6) is suggestive of nanogranular flow, which is not known for generating, or retaining a pre-existing, CPO. One potential mechanism for CPO formation in (nano-)granular flow may be through oriented interface attachment (OA), which is widely reported as a mechanism by which nanocrystallites can rapidly coalesce to form single crystals in numerous nanomaterials [160–162], including in calcite [163]. The thermodynamic driving force for particle coalescence in an OA event originates from crystallographic orientation-dependent, interatomic Coulombic interactions arising from both the surface atoms, and of atoms within the interior of the approaching nanoparticles [164,165]. Particle coalescence leads to a reduction of total surface energy [120], which, in the case of calcite would lead to an alignment of the lowest energy (104) plane [166], consistent with observations of simulated calcite faults deformed in LVF tests [113,114,140].

## 5.2. MSS-Bearing PSZs as Indicators for Past Seismic Slip?

Microstructural studies of simulated dolomite and limestone faults sheared in HVF experiments suggest the following characteristics of "glossy", "shiny" or "mirror-like" slip surfaces:

(i)   They form only at high mechanical work input rates or power densities ($\dot{W} = \mu \cdot \sigma_n^{eff} \cdot v$, where $\mu$ is the coefficient of fault friction) [135,138].

(ii)  They are at least in part responsible for the strong dynamic weakening often seen in samples sheared at co-seismic slip rates [137].

(iii) They are associated with dynamic recrystallization caused by heating at co-seismic slip rates [167].

However, the mirror-like surface patches formed in LVF experiments show very similar striated form and nanoscale topography [114,140] to those formed in HVF experiments, suggesting at least some degree of shared origin regardless of the areal extent or of shearing velocity. Moreover, a continuous MSS marking the PSZ has also been observed in simulated calcite gouge sheared at $0.1 \leq v \leq 100$ µms$^{-1}$ (Figure 5c), which showed steady-state $\mu$-values of ~0.5–0.6 (see [122,168] for data from experiments conducted under similar $T$-$\sigma_n^{eff}$-$v$ conditions). These observations strongly suggest that MSSs are not related to any dynamic weakening mechanisms, and that their seismic origin remains debatable at

best. Rather, as pointed out by Pozzi et al. [139], MSSs seem to demarcate a rheological discontinuity between an ultrafine-grained zone, which internally deforms via thermally-activated and grain-size dependent deformation mechanisms, i.e., the PSZ, and the adjacent, coarser-grained wall rock, which deforms by brittle processes (grain fracturing). The fact that patchy MSSs formed in LVF tests are found at different topographic levels throughout the PSZ (Figure 6a) indicate that they probably formed as isolated patches rather than as a single, through-going film. Furthermore, at some locations within the broader nanogranular PSZ volume, nanofibers are observed outside MSSs, or bridging between the MSS and the porous PSZ (Figure 6b,c). Combining these observations, we hypothesize that, with further increasing displacement, nanofibers within the bulk PSZ will ultimately align to form a single through-going MSS.

From rotary shear experiments on cylindrical cores of dolomite and limestone performed at $v$ = 0.002–0.96 m/s and $\sigma_n^{eff}$ = 0.25–6.9 MPa, Boneh et al. [138] showed that shiny striated slip-surface patches started to develop only at $\dot{W}$-values in excess of 30 kW/m$^2$. The cumulative area covered by these patches increased with increasing $\dot{W}$, ultimately producing a continuous, highly-reflective principal slip surface. Using experiments on simulated gouge prepared from dolostone, performed at $v$ = 0.001–1.13 m/s and $\sigma_n^{eff}$ = 13–26 MPa, Fondriest et al. [135] showed that shiny surfaces only developed at $\dot{W}$ values > 40 kW/m$^2$, covering an area of the sample that progressively increases with increasing displacement. By contrast, the shiny patches developed in simulated calcite gouge reported in LVF tests (Figure 6) [114,140] formed at $\dot{W} = \mu \cdot \sigma_n^{eff} \cdot v = 50$ MPa $\times (0.7 \pm 0.1) \times 10^{-6}$ m/s = 35 ± 5 W/m$^2$, i.e., 2 to 5 orders of magnitude lower than considered necessary for them to form in HVF experiments. This demonstrates that such shiny striates surfaces do not exclusively form at the high-power densities (>30–40 kW/m$^2$) associated with HVF experiments and with co-seismic slip rates. The implication is that mirror-like PSZs cannot be used as field indicators of past co-seismic slip in carbonates rocks without additional geological or microstructural evidence. The role of normal stress and cumulative displacement achieved in controlling the continuity of MSSs should be investigated further.

The development of highly-reflective PSZs in HVF experiments performed by Smith et al. [167], using simulated calcite gouge ($v$ > 0.1 m/s, $\sigma_n^{eff}$ = 2–26 MPa, $\sum x$ > 1 m) was shown to be associated with the presence of dynamically recrystallized grains characterized by a CPO, adjacent to the slipping zone, while the PSZ itself was composed of statically recrystallized grains. Dynamic recrystallization here refers to the growth of internal strain- or defect-free grains during shear, whereas static recrystallization refers to such growth upon piston arrest and cooling after the experiment. In the experiments by Smith et al. [167], recrystallization and CPO formation were attributed to the attainment of high temperatures (650–900 °C) reflecting heat dissipated from the PSZ during localized frictional slip at co-seismic rates [167,169]. Static recrystallization of PSZ grains was inferred to have played a role in experiments performed by Verberne et al. [122], on simulated calcite gouge sheared at $v$ = 100 μm/s ($T$ = 550 °C, $\sigma_n^{eff}$ = 50 MPa, $\sum x$ = 10.4 mm) (Figure 5d). Grain growth upon cooling after the experiment suggests that the grain size within the PSZ may have been smaller during shear. MSS-bearing PSZs developed in LVF tests ($v$ < 10$^{-4}$ ms$^{-1}$) using calcite gouge showed no evidence for conventional dynamic or static recrystallization, either in or adjacent to the PSZ, nor is this likely to have occurred considering the low slip rates, temperatures, and $\dot{W}$-values applying to these tests. The implication is that the presence of a statically recrystallized PSZ, with adjacent dynamically recrystallized grains, may indeed offer a useful constraint to past high-velocity slip, at least in limestones [167,170].

## 5.3. The Role of Nanocrystalline PSZs in Controlling Fault Stability

As mentioned in Section 2 above, in the case of earthquakes, fault sliding is believed to occur by stick-slip motion [38], caused by potentially unstable "velocity($v$)-weakening" properties of the fault sliding medium [39–41,43]. The physical processes responsible for $v$-weakening behavior of gouge-filled faults are only recently beginning to be elucidated. Based on observations from fault analogue experiments using powdered halite-muscovite mixtures, Niemeijer & Spiers [33,171] developed a micromechanical model for shear deformation of granular fault rock. They showed that

competition between dilatant granular flow and compaction by water-assisted diffusive mass transfer leads to an increase in steady-state porosity with increasing shear rate, and *v*-weakening behavior [171]. However, any time-sensitive, Arrhenius-type deformation mechanism will, when in competition with time-insensitive dilatation or granular flow, produce *v*-weakening. This mechanism of competitive dilatation and compaction may well explain thermally-activated transitions in the *v*-dependence of friction seen in a wide range of fault rock types [168,172–174]. The extended "Chen-Niemeijer-Spiers" (CNS) model developed recently [175] is capable of quantitatively reproducing a wide range of laboratory fault gouge friction data using physically-based input parameters [176,177], therewith providing a powerful tool for numerical earthquake-cycle simulators [178].

To produce *v*-weakening in the CNS model, the rate of intergranular compaction ($\dot{\varepsilon}_{cp}$) and dilatation by granular flow ($\dot{\varepsilon}_{gr}$) within the deforming gouge zone must be within the same order of magnitude, i.e., $\left|\dot{\varepsilon}_{gr}\right| \approx \left|\dot{\varepsilon}_{cp}\right|$. Under conditions where either process dominates stable *v*-strengthening occurs. In the case of intergranular creep by water-assisted diffusive mass transfer, relevant for compaction of microgranular calcite up to 150 °C [179], the compactive strain rate $\dot{\varepsilon}_{cp}$ is described using [72]

$$\dot{\varepsilon}_{cp} = A \cdot \frac{\sigma \Omega}{RTd^3} \cdot DCS \cdot f(\varphi) \tag{1}$$

where *A* is a constant, $\sigma$ is the (effective) axial stress, $\Omega$ is the molecular volume of the solid, *D* is the diffusion coefficient, *C* is the solubility of the solute, *d* is grain size, and $f(\varphi)$ is a porosity function (Table 3). In view of the inverse cubic dependence on *d* Equation (1), in the case of <100 nm sized particles in nanocrystalline PSZs, compaction by water-assisted diffusive mass transfer is expected to be very fast, even at relatively low temperatures.

**Table 3.** Values/expressions used for the terms appearing in Equation (1).

| Term | Formula/Value | Source |
|---|---|---|
| *A* | $576/3\pi \approx 61$ | Pluymakers & Spiers [180] |
| $\sigma$ | $50 \times 10^6$ Pa | Verberne et al. [140] |
| *d* | $1 \times 10^{-7}$ m | Figure 6 |
| *T* | 291 to 423 K | Verberne et al. [140] |
| $\Omega$ | $3.69 \times 10^{-5}$ m$^3$ mol$^{-1}$ | Zhang et al. [179] |
| $f(\phi)$ | $f(\phi) \approx 2\phi/(1-2\phi)^2 \approx 1.1$ | Pluymakers & Spiers [180] |
| Water-assisted diffusive mass transfer: | | |
| *D* | $D = D_0 \exp\left(-\frac{Q}{RT}\right)$ $D = 1 \times 10^{-10}$ m$^2$·s$^{-1}$ at $T = 298$ K $Q = 1.5 \times 10^4$ J·mol$^{-1}$ | Nakashima [181] |
| *C* | $C = \sqrt{K_s}$ $\log K_s =$ $-171.9605 - 0.077993T + \frac{2839.319}{T} + \log T$ | Plummer & Busenberg [182] |
| *S* | 1 to $2 \times 10^{-9}$ m | Verberne et al. [140] |
| Solid-state grain boundary diffusion: | | |
| *DS* | $DS = (DS)_0 \exp\left(-\frac{Q_d}{RT}\right)$ $(DS)_0 = 1.5 \times 10^{-9}$ m$^3$·s$^{-1}$ $Q_d = 2.67 \times 10^5$ J·mol$^{-1}$ | Farver & Yund [183] |

In the case of solid-state grain boundary diffusion *C* = 1.

In Figure 7 we plot Equation (1) for the case of granular calcite, assuming grain sizes in the range from 5 nm to 1 μm. We also indicate the conditions of temperature (25–150 °C) and intergranular dilatation rate ($\left|\dot{\varepsilon}_{gr}\right| \approx 10^{-2}10^0$ s$^{-1}$) characterizing the PSZ developed in LVF experiments on simulated calcite faults (Figure 4a,b and Figure 6) [140]. This shows that for 100 to 500 nm-sized grains $\left|\dot{\varepsilon}_{gr}\right| \approx \left|\dot{\varepsilon}_{cp}\right|$ (Figure 7a), implying that under the conditions of normal stress and temperature used here (Table 3) *v*-weakening may be observed. Combined with the ~100 nm-sized nanospherules and -fibers observed in the broader

PSZ (Figure 6), this suggest that the mechanism of competitive dilation and compaction [33,171] can explain the thermally-activated transition from stable *v*-strengthening to unstable *v*-weakening seen at ~100 °C in experiments on simulated calcite(-rich) fault gouge [114,124,168,184,185]. For grains <100 nm size, this mechanism is not expected to be relevant when assuming intergranular creep by water-assisted diffusive mass transfer. However, equation 1 can also be used to assess creep rates assuming solid-state diffusion involving mass transfer through a grain boundary of thickness *S* [70]. This shows that for grain sizes down to 10 nm, $\dot{\varepsilon}_{cp} \approx 10^{-3}$ s$^{-1}$ at temperatures > 650 °C (Figure 7b). Such high temperatures are common in HVF experiments, suggesting that the mechanism of competitive compaction and dilation may also play a role at co-seismic slip rates. This is consistent with claims that superplastic deformation of nanogranular fault rock controls (dynamic) fault rupture [93,116,186].

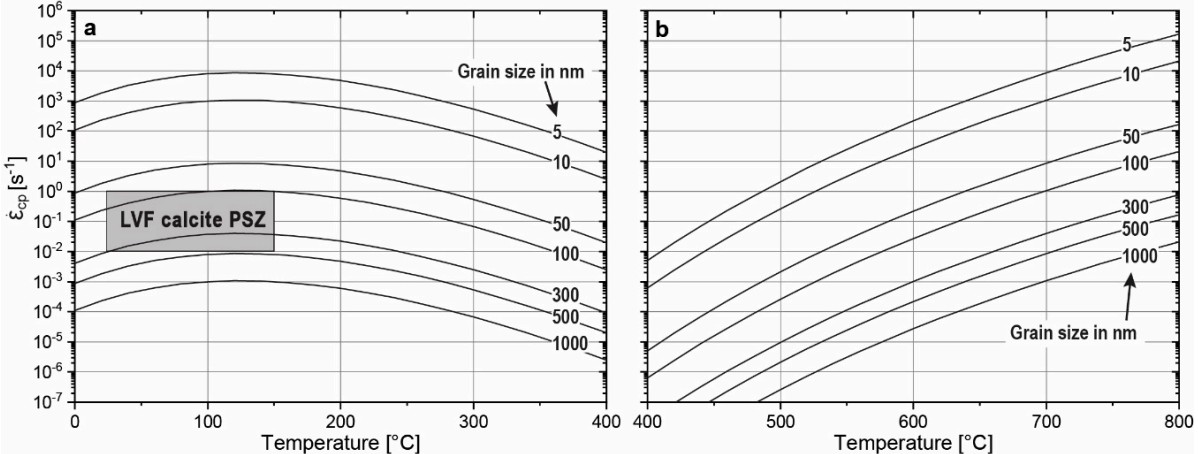

**Figure 7.** Intergranular compaction creep strain rates in granular calcite versus temperature for grain size *d*. (**a**). Water-assisted diffusive mass transfer. The grey shaded area indicates the conditions characterizing the PSZ developed in LVF experiments using simulated calcite gouge (Figure 4a,b and Figure 6) [140] (**b**). Solid-state grain boundary diffusion. For a list of parameters used see Table 3.

Notwithstanding all of the above, the micromechanical framework underlying the CNS model [171,175] is of course highly idealized. The model assumptions are reasonable at low slip rates, but break down when frictional heating and associated dynamic fault rupture processes come into play [57,61,62]. Another potentially problematical aspect is the knowledge and quantification of the relevant intergranular creep mechanisms. In view of the unusual deformation properties of nanocrystalline materials (Figure 2), extrapolation of data from compaction experiments using microcrystalline samples to the nanometer realm may be unreasonable. Parameter values and expressions such as listed in Table 3 have to be re-assessed in the case of nanocrystalline fault rock.

### 5.4. Implications for Natural Faulting in the Seismogenic Zone

In the above we have shown that the Chen-Niemeijer-Spiers (CNS) model describing competitive dilatant nanogranular flow and nanospherule/-fiber compaction may explain the transition from stable velocity strengthening to potentially unstable velocity weakening at temperatures ~80–100 °C seen in calcite fault rock [114,124,168,184,185]. This transition is consistent with the location of the upper seismogenic limit at shallow depths (~2–4 km) in tectonically-active limestone terrains, such as those characterizing the Mediterranean region [187–189]. Velocity weakening hence seismogenic fault slip on nanogranular PSZs becomes possible because (water-assisted) diffusive mass transfer is dramatically accelerated by the nanogranular nature of the slip zone rock that forms. Given the abundant observations of nanogranular fault surfaces in fault rocks of all types (Tables 1 and 2), and the fast diffusion rates in nanostructured materials [11,69,75], the mechanism of dilatation versus

compaction, applied to sheared nanogranular fault rock, may be generally relevant in controlling the upper limit of the seismogenic zone.

In the CNS model framework, a comparison between fault rock creep rates and fault zone shear strain rates may yield clues on the depth to the limits seismogenic zone, i.e., taking the condition that $|\dot{\varepsilon}_{gr}| \approx |\dot{\varepsilon}_{cp}|$ may lead to unstable seismogenic fault-slip [33,140,171]. To illustrate this, we used Equation (1) for the case of calcite (Table 3) to plot grain size vs. depth curves for different $\dot{\varepsilon}_{cp}$, assuming a geothermal gradient of 30 °C/km and a density of 2700 kg/m$^{-3}$ (Figure 8). Crustal fault zone shear strain rates remain poorly constrained, mainly because of the lack of observations on the width of the actively deforming zone [190]. Nonetheless, using the "commonly cited value" for upper-crustal fault zone shear strain rates of ~10$^{-14}$ s$^{-1}$ [190,191] for $\dot{\varepsilon}_{gr}$ in Equation (1) (Figure 8), shows that solid state creep may be relevant in grains <100 nm in size at ~3–4 km depth. This is close to the upper limit of the seismogenic zone in tectonically-active carbonate terrains [187–189]. However, importantly, in the above analysis using the compaction-dilation model, the processes controlling nanograin formation and the role of grain growth are ignored. Especially the latter potentially presents a major limitation, since at greater depths/temperatures static as well as dynamic recrystallization of PSZ grains are expected to play a role [147,192]. Future models aiming to describe the physical processes leading to dynamic fault rupture should take into account the progressive development of fault rocks with increasing shear strain, i.e., specifically, the competition between grain growth and grain size reduction the PSZ.

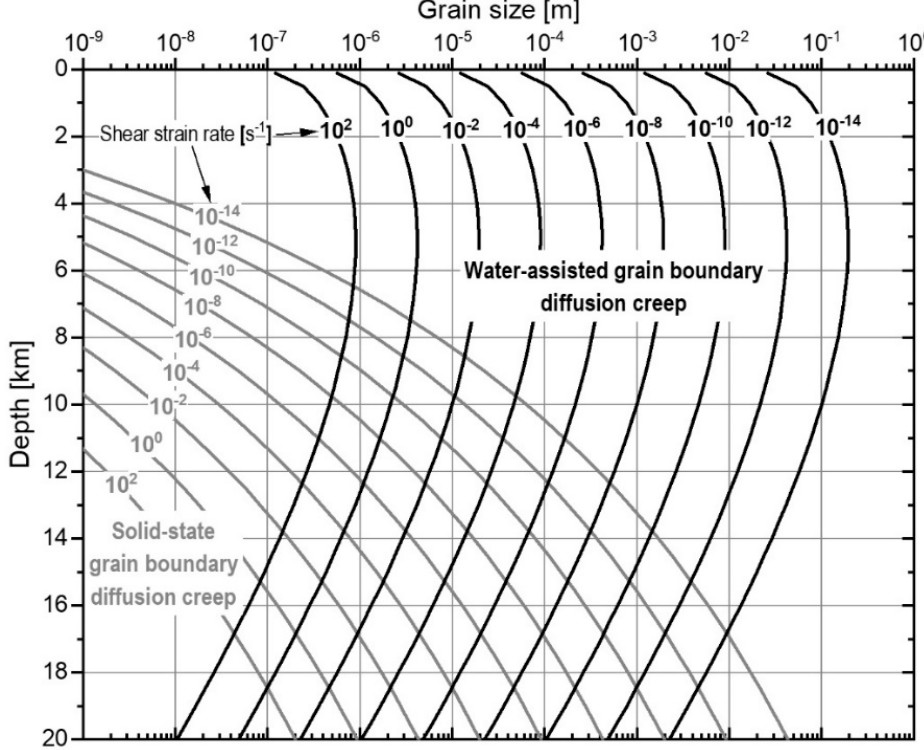

**Figure 8.** Water-assisted and solid-state grain boundary diffusion creep in calcite (Equation (1)), plotted as grain size vs. depth (taking 30 °C/km and a density of 2700 kg m$^{-3}$). For a list of parameters used see Table 3.

## 6. Conclusions

Nanocrystalline fault rock is consistently observed in natural and experimentally-formed principal slip zones (PSZs) and is frequently associated with the presence of a crystallographic preferred orientation (CPO), (partly) amorphized materials, and ultra-smooth interfaces known as "glossy", "shiny" or "mirror-like" slip surfaces (MSSs). Experiments conducted under a wide range of normal

stresses, temperatures, and displacement rates demonstrate that these features can be produced over a wide range of conditions pertaining to upper-crustal fault-slip, covering co-seismic and sub-seismic displacement rates. Simple calculations using constitutive equations for compaction by water-assisted diffusive mass transfer, combined with existing models for velocity-weakening shear of gouge-filled faults, show that nanogranular fault rock plays a key role in controlling the depth to the upper-limit of the seismogenic zone. In view of the unusual deformation properties of nanocrystalline (NC) materials, an important task in Earth sciences is to improve insights on the rheology of NC PSZs, or of nanophased geomaterials in general.

**Supplementary Materials:** The following are available online at http://www.mdpi.com/2075-163X/9/6/328/s1, Video S1: unif_biref-[CaCO3-RT-dry].

**Author Contributions:** Conceptualization, B.A.V. and O.P.; methodology, B.A.V.; software, N/A; validation, B.A.V., O.P. and C.J.S.; formal analysis, B.A.V.; investigation, B.A.V.; resources, C.J.S.; data curation, B.A.V.; writing—original draft preparation, B.A.V.; writing—review & editing B.A.V., O.P., C.J.S.; visualization, B.A.V., O.P.; supervision, C.J.S.; project administration, B.A.V.; funding acquisition, B.A.V., C.J.S.

**Funding:** Part of this work was conducted within the framework of B.A.V.'s Ph.D. thesis, which was supported by grant 2011-75 awarded by the Netherlands Centre for Integrated Solid Earth Science (ISES). B.A.V. is now supported by JSPS KAKENHI grant #19K14823.

**Acknowledgments:** The authors thank André Niemeijer, Hans de Bresser, Jianye Chen, Colin Peach, Virginia Toy, and Martyn Drury for discussions at various stages of this work. Matthijs de Winter is thanked for preparing the FIB-section in sample lmst@150 °C (Figure 4b). Michael Hochella, Li-Wei Kuo, and Hiroko Kitajima are thanked for helping with literature. We thank two anonymous reviewers for helpful input that improved the paper significantly.

**Conflicts of Interest:** The authors declare no conflicts of interest.

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
