# Peer review of "Nanocrystalline Principal Slip Zones and Their Role in Controlling Crustal Fault Rheology"

_minerals, doi:10.3390/min9060328_

Round 1
Reviewer 1 Report
See attachment

Author Response
See .pdf file attached

Reviewer 2 Report
This is a very good article that has two basic parts; first a summary of observations of principal slip zones consisting of nano-scale particles both in the field and in the laboratory, and secondly a discussion of mechanisms regarding their formation and how the presence of nano-particles may influence the occurrence of seismic slip. There is a particular focus on shiny, mirror-like surfaces in calcite. The paper is well written and informative, and should be published with a few minor revisions.
Comments:
It is true that nanograin PSZs form under a wide range of conditions, but I think people are interested in whether they form under seismic conditions or if they can also form under aseismic conditions. The answer is clearly both, which is a point that maybe could be made more prominently (in the abstract, for example).
I found the paragraph on Lines 209-219, especially the last sentence, especially intriguing with potentially huge implications for identification of past earthquake slip. I would like to see this paragraph expanded, including explanations of what exactly static and dynamic recrystallization are and discussion of the conditions under which each may or may not occur.
The discussion in Section 4.3 and 4.4 are based on the assumption that the nanograins already exist, regardless of their formation, right? I think this point should be made more clearly at the beginning of this discussion. Also, this analysis shows that nano-scale grains can deform fast enough to be considered seismic at “seismogenic depths” – but then shouldn’t the deformation always be seismic since the P-T conditions are constant? In other words, what would be the mechanism for seismic and interseismic cyclicity? What about the possibility that the nanograins can agglomerate back into larger grains? I did not see this discussed but this could induce cyclicity.
There appears to be text missing or typos at Lines 65, 137, 151, and 260-261.
Author Response
See .pdf file attached
